# The ABC Model of Happiness—Neurobiological Aspects of Motivation and Positive Mood, and Their Dynamic Changes through Practice, the Course of Life

**DOI:** 10.3390/biology11060843

**Published:** 2022-05-31

**Authors:** Tobias Esch

**Affiliations:** Institute for Integrative Health Care and Health Promotion, School of Medicine, Witten/Herdecke University, 58455 Witten, Germany; tobias.esch@uni-wh.de

**Keywords:** motivational salience, reward, U-curve of happiness, life satisfaction, subjective well-being, aging, mindfulness, contemplative practice

## Abstract

**Simple Summary:**

This article proposes a new model for exploring happiness primarily from a neurobiological perspective. Such understanding includes the dynamics of positive mood states and how they change throughout life. Happiness is not a cognitive construct: it is an immediate emotional experience—a feeling that relies on neurophysiological activation in the brain’s reward system. With this in mind, three types of happiness are proposed: (A) *wanting*, approaching, and pleasure, (B) *avoiding*, departing, and relief, (C) *non-wanting*, staying, and satisfaction. Behind this is a sophisticated (neuro)biological dynamic, ranging from the search for autonomy and ecstasy, which is particularly characteristic of young people, to the way we cope with stress, as we find it pronounced in the middle-aged, to deep contentment, peace, and inner joy, as it is mainly attributed to older people. Paradoxically, it is in fact the elderly who appear to be the happiest and most content—this phenomenon is also known as the “satisfaction paradox”. Apparently, these dynamic changes in happiness can be amplified with practice. Happiness is biological in this context, but can still be “learned”. Contemplative practices can serve as an example here to demonstrate this trainability, and they may themselves influence the course of happiness.

**Abstract:**

Background: Happiness is a feeling, an immediate experience, not a cognitive construct. It is based on activity in the brain’s neurobiological reward and motivation systems, which have been retained in evolution. This conceptual review provides an overview of the basic neurobiological principles behind happiness phenomena and proposes a framework for further classification. Results: Three neurobiologically distinct types of happiness exist: (A) wanting, (B) avoiding, and (C) non-wanting. Behind these types lies a dynamic gradation, ranging from the more youthful anticipation, pleasure and ecstasy (A), to stress processing, escape and relief (B) as we find them accentuated in the middle-aged, to deep satisfaction, quiescence and inner joy (C), which is particularly attributed to older people. As a result, the development of happiness and satisfaction over the course of life typically takes the form of a U-curve. Discussion: The outlined triad and dynamic of happiness leads to the paradoxical finding that the elderly seem to be the happiest—a phenomenon that is termed “satisfaction paradox”. This assumed change in happiness and contentment over the life span, which includes an increasing “emancipation” from the idea of good health as a mandatory prerequisite for happiness and contentment, can itself be changed—it is trainable. Conclusions: Programs for mindfulness, contemplation, or stress reduction, including positive psychology and mind–body/behavioral medicine training, seem to be capable of influencing the course happiness over time: Happiness can be shaped through practice.

## 1. Introduction

Research on happiness, especially with regard to its psychosocial as well as biological implications or its differentiation from satisfaction have gained momentum in recent years [1,2,3]. Today, happiness can be viewed as a highly relevant phenomenon: Happiness “makes sense” from a biological point of view and is not restricted to humans in principle [4]. Hence, happiness is not “intellectual”; it is not constructed or cognitive. Primarily, it is a feeling, not a thought, and it is also physical at the same time, that is, based on the motivational and reward system inside the brain [5,6]. Here, cognitive or evaluative parts are downstream, i.e., secondary or associative in nature [7].

The biology of happiness follows rather objective criteria. There is an overarching goal (e.g., survival, reproduction, behavior control [7,8]) that, if necessary, can override subjective or individual goals. Ideally, however, the primary (objective) and the secondary (subjective, subsequent) goals are aligned, that is, well-coordinated with one another. With regard to these secondary parts, study participants give different statements in surveys on the subject of happiness: When asked about their secondary assessment of happiness, respondents give answers that frequently diverge from those who had been queried about their primary feelings, i.e., about the direct experience in actual (happy) moments (c.f., subjective well-being (SWB) as a cognitive construct in contrast to the current emotional state or the mood in any given, immediate moment) [7,9,10].

Hence, happiness is not just a “good feeling”. Much more, it carries along strong biological cues and values. It provides inner guidelines, i.e., directions, and lets us humans plan or implement behaviors from which we hope to derive a desired (beneficial) outcome [2,5,6,8]. Therefore, the feeling of happiness—if based on an experience that has already been made before (i.e., memorized, conditioned, and also linked to it: a positive expectation for the future)—is an efficient way of integrating or automating experiences made earlier into behavioral or treatment concepts that can be quickly implemented and activated (i.e., short-cut) [8,11,12].

Happiness as a biological concept was passed on genetically and further developed in evolution [4,7,13,14,15]. It has a biomolecular basis yet is used for autoregulation and survival (of the individual and of the species) [5,7,12,16,17]. Happiness can thus be measured (e.g., in the brain, blood; see below) and is associated with biophysiological changes in the body [7,18,19,20,21,22,23,24]. At its core, happiness, as is the case with the entire body (the living organism), is dynamic by nature and is subject to cyclical or internal “maturation processes” (see below). Thus, an interesting question coupled to this is whether happiness or its trajectories can actually be trained, i.e., actively be altered, for example, through positive psychology programs, contemplative or meditative practices.

To summarize, we define happiness as a natural, biologically “meaningful” phenomenon—a positive, pleasurable, or rewarding state that makes us feel, i.e., emotionally realize, an inner “calling”: a confirmation to show us what is supposedly good or biologically beneficial for us—in this moment or over time. Happiness thus (invisibly) controls our behavior. Our SWB is based on this, yet subsequently dependent on it.

Happiness is therefore neurobiologically “produced” in the brain by the reward and motivation systems. We surmise that each of these “happy” mood states has a specific representation—i.e., a corresponding, analogous activation pattern—in the brain (see below). Since these states (patterns and pathways) are also related, e.g., to addiction or other external phenomena (i.e., drugs) [5,7], there can be negative, less beneficial interactions with behavior regarding the fact that these do not necessarily always have to be medically healthy (see below). This applies in particular to sub-forms of happiness, such as those associated with ecstasy or wanting (type A), or those directly related to stress (type B)—as it will be described in the next section.

Hence, the “hardware” for happiness is innate (albeit variable, in moderation), yet the “software” or operating systems are regularly updated through their use: what makes us happy is individual (to a large extent), but the structures and functions seem universal.

As we will see later in this conceptual narrative review, happiness changes its “color” over the lifetime: from desire and wanting, to relief and stress reduction, to satisfaction and non-wanting, possibly “inner peace”. In this respect, we understand happiness on the one hand as an umbrella term for the various subtypes of “happiness” (motivation- and reward-related positive mood states, SWB)—that also change over a lifetime—but on the other hand as a more specific component of the (sub-) state, which is particularly associated with inner joy and contentment (type C; see below).

## 2. Three Types of Motivation and Rewarding (“Happy”) States

As stated above, happiness describes a feeling that is manifested biophysiologically in the brain, mind, and body. The feeling that comes along with it serves to memorize its initial occurrence and its antecedent, i.e., the contexts that led to or accompanied it. By this mechanism, “happy events” not only become conditioned and reinforced (i.e., learning and wanting to experience them again) yet they also become stored in memory together with an emotional “tag” for better and faster retrieval of the stored information later on, i.e., when it is biologically required or circumstances trigger memories of the initial event [7].

As with happiness, behaviors (and so: lifestyles, health behaviors) are shaped by implicit emotions and autonomous, unconscious processes (i.e., non-cognitive motives) rather than by metacognition or cognitive willpower (e.g., see [8,25,26,27,28,29]). In fact, health behaviors that are experienced as pleasant are more likely to be repeated (e.g., [5,30,31,32,33]). To better understand the underlying mechanisms of reward and behavioral motivation, including their neurobiological significance, I suggest distinguishing between three types of motivational states, namely, (A) approach motivation, (B) avoidance motivation, and (C) assertion motivation (Figure 1). The rationale behind this distinction and their implications will be explained in the following.

Motivation describes processes that represent the core of biological, cognitive, and social regulation [5], including the regulation of intensity of behavior that leads to the attainment of a particular goal or stimulus [34]. We define stimuli as concrete physical objects, mental representations or memories of such objects, abstract concepts, or possibilities that are expected to occur in the future [35,36,37]. The pursuit (effect) of stimuli is based on affects (or “basic emotions”, see [38,39]) and can be automatic and occur with or without awareness. When cognitively processed, we may call these stimuli “goals”. Thus, behavior can be stimulus-driven (affective) or goal-directed (cognitive) [40], with the prior occurring most often (e.g., [8]).

Motivational salience is the attribute of a stimulus and can be appetitive, aversive [34], or assertive [7,41]. However, the existing literature does not sufficiently differentiate between assertive and appetitive motivation; usually, both fall under the umbrella term “approach motivation”. As noted earlier, I now suggest distinguishing three types of motivational processes (ABC), since they involve different physiological/neurobiological mechanisms (Figure 1).

### 2.1. Approach Motivation

Approach motivation, or appetitive (incentive) salience, is directed towards stimuli or goals that are related to positive, hedonic, pleasant processes [5,36,42] and functionally linked to the wanting system, i.e., reward expectation, performance and action [7]. This type of motivation is sometimes also referred to as “wanting motivation” [7].

If approach motivation (appetite, wanting) has led to the achievement of a stimulus or goal, a reward is experienced as a pleasant feeling (which, depending on the intensity of the experience, can also go unnoticed). The stimulus (goal) itself does not serve as a reward. Instead, reinforcement occurs either via psychological and related neurobiological processes that take place during one’s anticipatory (expectant) state and/or as a response to the actual stimulus or goal [43,44].

Individuals continually evaluate stimuli and determine them to be either beneficial or detrimental [36]. These evaluations are often perceived as basic affective experiences [36,45,46,47,48]. Simply put, approach motivation emphasizes the expectation of a reward in form of pleasant feelings, or positive affect, e.g., joy, pleasure, and excitement.

Underlying the concepts of motivation are physiological mechanisms that occur in brain areas distinct from other sensory or cognitive areas [5,7,49]. An integral part of the central nervous system (CNS), approach motivation and reward systems are neurons that have their principal origin in the ventral tegmental area (VTA), located in the midbrain [5,12]. These neurons send projections, e.g., to the frontolimbic brain, mostly to the nucleus accumbens (NAcc) [50,51]. By this, the midbrain and the frontolimbic systems are “wired” together; this is known as mesolimbic coupling. When a stimulus is paired with a reward initially or when a reward is anticipated in response to a stimulus based on past experience, via the mesolimbic pathway with its essential neurotransmitter dopamine, VTA and NAcc become connected and activated [52,53], i.e., “dopaminergic activation promotes positive feelings” [7]. The NAcc hence signals the desire to obtain a reward and the degree of effort to do so, i.e., it determines appetitive motivational salience. In addition, reward is also measured and regulated by the VTA-NAcc pathway—it signals to other brain areas how rewarding an activity is [5]. The intensity of an expected reward, consequently, determines how likely it is that a person will remember and repeat it [12]. Here, the hippocampus, as part of the (para)limbic system, serves as the point of entry for experiences to be recognized and remembered [5,54].

Additionally, by also involving the amygdala, our brain categorizes and remembers experiences as pleasant or detrimental within the endogenous reward system, which in turn facilitates the pairing between experiences and other stimuli [12,50,51]. Such information is also processed via the mesocortical (mesofrontal) dopamine pathway in the frontal cortex, which is integral to weighing one’s personal “costs” (disadvantages) against the likelihood of reward. Whether the behavior is ultimately performed or not is determined by these dynamic processes of deliberation, with the hedonic value driving the wanting, i.e., the appetite and approach motivation [8,12].

### 2.2. Avoidance Motivation

Aversive motivational, or negatively-valenced “fearful” salience, is related to the avoidance of pain or threat (i.e., threat avoidance) and punishment (i.e., consequences that diminish the chance of the reoccurrence of a behavior), and it corresponds with the fight-flight-freeze system (i.e., stress physiology, stress response) [5,7,42,55]. It is usually triggered by an aversive stimulus and is motivated by a desire to attain relief from undesirable circumstances.

Punishment, which is also linked to a reduced response strength (i.e., passive avoidance), can further be distinguished from negative reinforcement, the latter of which is related to an increased response strength (i.e., active avoidance) [44]. Thus, in comparison to active fear reactions (e.g., fight or flight) resulting from fearful stimuli, passive responses (e.g., freezing) can also occur [56].

Anxiety, fear, and disgust are examples of negative affects associated with avoidance motivation [7,36,45,57,58,59]. Functionally embedded in the stress system, avoidance motivation is related to increased sympathetic and stress activity (i.e., activated stress responses), including the release of cortisol, (nor-) adrenaline, as well as, e.g., opioids and vasopressin [7]. Anatomically, it is rooted in the lower limbic system, primarily in the amygdala and hypothalamus (connected to the pituitary gland) [12].

Two central signaling pathways are activated when threat is anticipated: One pathway involves the hypothalamus, which receives its impulses from cortical, subcortical, limbic and brainstem areas, and leads to the pituitary gland, where (pre-) hormones are released into the blood, triggering the release, e.g., of cortisol from the adrenal cortex. Cortisol is particularly responsible for providing energy for “fight and flight” [7]. The other pathway acts via the sympathetic autonomous nervous system. Here, too, the hypothalamus, but above all the brainstem and the brain’s catecholamine systems, are crucial. From here, nerve impulses are directly (neuronally) transmitted to the peripheral organs and the adrenal medulla. The stress hormone adrenaline (or noradrenaline) is released from there. This second pathway is the faster, more direct one, and particularly affects the circulatory and organ functions [7,12]. In addition, the freeze response, as mentioned above, is functionally embedded in the CNS amygdala circuitry [60].

In the context of successful passive or active threat avoidance, the modulation of the various signaling pathways involved ultimately leads to the perception of relief, which is a positive, low-arousal basic emotion/affect [61,62], corresponding with one’s original (conscious or unconscious) intention, i.e., to survive. Indeed, when the aversive stimulus is discontinued, actively or passively, relief occurs [63]. Psychologically, relief results from the reduced impact of a negative stimulus and can be experienced as relaxation and/or reward, i.e., “no more stress” [6,12,64,65].

Relief is also experienced when the activity of the amygdala is reduced [7,12]. This, too, signals (or is the biological consequence of) less or “no stress”—which, again, could be an active or a passive process. Furthermore, we see a connection between approach and avoidance motivation systems here: Results, e.g., from functional magnetic resonance imaging (fMRI)-based studies suggest that corresponding brain areas are activated during relief and other positive affects (e.g., see [66,67]).

### 2.3. Assertion Motivation

As described above, the majority of previously published research on motivation and reward does not distinguish between behavior driven by approach versus assertion motivation. In fact, these constructs or states are often confounded or merged with each other [41], even though they reflect different neurobiological processes, are originating from distinct areas of the brain, and have different behavioral consequences [7,8]. Assertion motivation, or assertive salience, is associated with the “non-wanting” system or “non-wanting motivation”, hence with inaction (i.e., staying), acceptance or contentedness, and quiescence. It characterizes the motivation to maintain a certain condition or state [7,41]; the associated positive valence is contentment. The assertion motivation differs from the approach motivation with regard to the affective qualities involved [7,41]. Furthermore, assertion motivation can also be distinguished from approach motivation in terms of different automatic responses and behavioral outcomes.

Assertion motivation is based on the lack of a goal-directed, recognizable action (neither having to get somewhere nor having to get away), because one has purposefully (explicitly) or unconsciously (implicitly) consented to remaining in the current state, for example, a newly habituated health behavior. Consenting to the current state can be experienced as being “perfectly happy”, or content, having no intention to change it or move away from it, or shift one’s attention towards some other thing or place. Hence, by being in congruence with or in full acceptance of the present moment, this state of non-wanting also entails a feeling of “mindful” connectedness (an alignment with the “here and now”), and the underlying system is thus sometimes called the “affiliation system” [7]. Unsurprisingly, mindfulness or meditation practices seem to facilitate this kind of experience [68,69].

Functionally embedded within the parasympathetic autonomous nervous system, the assertive motivational state is linked to increased parasympathetic or vagus nerve activity and is therefore associated with a physio-psychological down-regulation, and states of relaxation [70,71,72,73]. At the level of neurotransmitter systems, assertive salience is associated with endogenous opiates, oxytocin, acetylcholine, serotonin, as well as endocannabinoid signaling [7,12,20,69,74,75,76]. Unlike approach and avoidance motivation systems, the assertion motivation system is not characterized by an involvement of dopamine—instead, it is linked to the absence or inhibition of dopaminergic activation. Thus, individuals here experience no motivation to change the status quo by generating or avoiding new experiences through behavior change. Brain areas involved in the activation of assertive motivation include, but are not limited to, the midbrain, the vagus areas, brainstem, cingulum, hippocampus and ventral striatum, as well as the hypothalamus and the pituitary gland [7].

## 3. How Motivation and Happiness Change over the Course of Life

The insufficient distinction between the constructs of approach and avoidance motivation (i.e., motivated behavior; see above, Figure 1) underscores the need for further operationalization of the constructs, but also for the addition of a third dimension, namely assertion (affiliation). The latter has been neglected in the literature published to date, as has another relevant finding that describes a crucial piece of the puzzle, namely the changes in happiness, motivation or satisfaction that occur over the course of a lifetime.

The adoption of our ABC model of reward and motivation and its relationship to happiness mentioned above began with observations from basic research. For example, we could identify pathways by which dopamine is formed in neuronal tissues, so we could map the first stage (A) of happiness, and then we could show how endogenous opiates, for example, are ultimately formed in stage C by involving dopamine metabolism (supported by stress hormones that characterize stage B) [73,77,78,79,80,81,82,83,84,85]. Thus, the modeling of a “neurobiology of happiness” was initially not based on empirical or clinical data (as would be common practice in the medical field) but instead originated from theoretical considerations, i.e., deductively. Specifically, the model has been derived from the knowledge of which neurotransmitter first appeared in evolution, and which metabolic pathways were connected with one another [83]. From this notion, the assumption was derived that we would expect level A happiness (type A salience) predominantly in early or “youthful” times of life; level B, however, in middle phases of life; level C, for example, primarily at an older age [7].

Initially, these considerations were purely theoretical. We were uncertain whether they could be transferred from (neuro-) biological evolutionary processes (i.e., phylogenetic observations) to the ontogenetic development of an individual in the course of life. However, these considerations inevitably led to the assumption that happiness, especially with regard to the aspect of growth (maturation, thriving), wanting, and anticipation (type A)—that is, the momentary and more ephemeral form of happiness (i.e., pleasure), which would be “ecstatic” and expressed with “high moments” (peak moments: short-lived and transient)—should be more pronounced in youth, whereas the middle phase of life would more likely (at best!) be characterized by relief and the “absence of unhappiness” or absence of threat (B), while later phases, in the elderly, would in turn be characterized by a rather lasting “inner joy” and contentment (C).

In short, we anticipated happiness to fluctuate during individuals’ lifetime, reflecting a “U-curve”. Given that old age has not been associated with happiness and contentment in the past [86], we initially found the notion of a U-curve of happiness to be paradoxical.

Only by gaining direct insight into large longitudinal clinical (empirical) data sets, for example, the Nurses Health Study [87,88] or the Grant Study (Harvard Study of Adult Development) [89], as well as the UK Million Women Study [90], and now many more, have we been able to confirm the assumptions derived from our theoretical model: In fact, “old age” seems to be characterized by higher levels of life satisfaction (statistically, except for the very last phase of life, the so-called “fourth age”) [91]. In the meantime, current findings show that we can find such a U-curve of happiness in almost all populations and countries around the world [92]. Our own current data confirm this finding among the German population (see Figure 2 and [9,10]).

What do we actually mean by “age” or the “elderly”? The classification of the various life and aging phases differ depending on the discipline and specific perspective. According to World Health Organization (WHO) and current gerontology frameworks, the chronological age limit of 65 years marks the beginning of seniority (i.e., the “elderly”) [93], followed by further phases and subdivisions, depending on expert opinions: One speaks of “young age” and “old age”, whereby the onset of old age is usually assumed to be 80–85 years [93]. This later phase, equivalent to the fourth age (see above; [91]), is also the phase when health impairments, the need for help and care, and ultimately mortality, increase [93]. Some authors distinguish from this again a phase of the very old or “centenarians” [94]. For pragmatic reasons, we ourselves correlated “the elderly” in our analyses (e.g., [9]) with the statistical second half of life (cf. [86]), thus beginning about 10 years earlier than in the gerontopsychological literature [93,95,96], but otherwise adhere to the above-mentioned classification of ages.

Hence, as indicated, the neurobiological model of motivation systems could help explain the observed pattern of a U-curve, with the elderly experiencing highest levels: According to this, life satisfaction and happiness result from lifelong maturing processes driven by constant endogenous rewards and motivation cycles. Neurobiological processes shape our maturation by chemically and biologically rewarding the “right” practices and associated experiences, i.e., trajectories of a “good” or “fulfilling” life.

As described, the model distinguishes three levels of motivation (A–C), all of which share the common aim of advancing personal and biological growth throughout the various phases of life. After starting life with extensive freedom, a vast adaptation potential, and a brain that remains inadequately prepared for the concrete challenges of life, we need to learn and adapt our inner structure to the outer world (neuroplasticity). During this first phase, we are biologically adaptable. We want to evolve, thrive, learn, explore the unknown, and constantly interpret and change in response to the various impressions that life (i.e., the external world) has to offer (type A motivation: the “wanting system”). With progressive adaptation and maturation, our stress physiology is activated, and we strive to defend what has been gained (and has been ingrained in our structure) rather than conquer new territories. We long for security and protection (i.e., survival) and want to avoid harm and anxiety (type B motivation: the “threat avoidance system”). If we succeed in adapting to our environment and maturing (thriving) internally, the development of deep and persistent satisfaction is possible (type C motivation: the “non-wanting” or “quiescence system”). Happiness becomes a way of being; an internal state (Figure 3).

As part of the model, also following suggestions from other researchers (e.g., [97]), I propose a distinction between momentary happiness and life satisfaction. While momentary happiness is characterized by intense, pleasurable, and euphoric but fleeting moments, life satisfaction is more profound, persistent, and subtle, and it is characterized by feelings of acceptance, affiliation, arrival, and quiescence, among others [7] (see also Figure 1). In this context, again, it is important to note that satisfaction, too, is not a thought or a thought construct (a cognitive concept) but rather a feeling and a deep emotion—a lasting inner state. Particularly, the stress experienced in phases A and B of life appears to be a prerequisite for this deep and prolonged satisfaction later, i.e., in phase C (Figure 4).

The increase in life satisfaction observed in the second half of life may seem counterintuitive, as this period is often characterized by increasing physical complaints and the onset of chronic diseases. Consequently, social researchers and gerontologists have called this phenomenon a “satisfaction paradox” (e.g., [86,98,99,100,101]). Our neurobiological model, however, could explain why the increase in satisfaction observed might be less paradoxical than commonly assumed. As we age, our needs and desires are satisfied in a more “cost-effective” and efficient manner. This is perhaps owed to our brain’s ability to constantly adapt to the ever-changing demands of our social environment [7]. It seems that we get better and better at it over lifetime. Moreover, the model could also explain the observed increase in satisfaction *above* baseline: not only are expectations met more efficiently, and one is satisfied with what is available (i.e., successful “expectation management”)—or what is still possible—but one can obviously experience greater satisfaction than ever before, which indicates personal growth and flourishing. Indeed, as analyses have shown, life satisfaction begins to decline only a few years prior to death, e.g., caused by prolonged illness [102,103]: the fourth age (as described). Before the onset of this stage, major or severe depression is less common among older than younger individuals [86], whereas life satisfaction is commonly high.

As one reason for this finding, from an evolutionary point of view, we assume that there is a movement over the lifetime from the self (ego), and “me” becoming (which also includes taking risks), to being, preserving, and procreating (for this, one also needs to activate their defensive shields, i.e., fight or flight: accepting stress, being able to cope with it), towards, finally, letting go, which also includes a “loving acceptance”, and transcendence (cf. [89]). In the end, one has to be ready to leave this life again, finding satisfaction and “inner peace” with it, and be able to pass something on, be a role model. The latter is also called wisdom or generativity, and it has an important sociobiological function [89,104,105]. In this aspect, the “we” (as opposed to the “me”) and a deep, cross-generational connection are expressed, as evidenced also by neurobiological findings (e.g., on oxytocin, with higher levels in old age [20,76], or on generosity and happiness [21]).

As with some other authors (e.g., [92,106]), we observed a decrease in life satisfaction from early adulthood until midlife, especially for male participants [9,10]. According to our neurobiological model, early adulthood is a phase in which most people experience the seriousness and challenges of life (i.e., stress). At this stage of our lives, we need to assume responsibility for ourselves, our children, and sometimes also our parents or other people relying on our care. We usually work more and start feeling the burden of financial pressures. Consequently, the two stress axes originating in our brain activate the body’s physiological stress response (“allostatic stress response” [71]); we are alert and ready to fight the challenges and difficulties of life [5,12,107,108]. While sometimes biologically necessary, the consequences of an uncontrolled stress response, or simply too much stress, can cause life satisfaction to stall or even decline until the brain adapts, learns, and life satisfaction begins to rise again [71].

## 4. Can Happiness Be Learned? Long-Term Effects of Stress Reduction, Contemplative Practice

We have seen that happiness and contentment (satisfaction) change over a lifetime—in their quality, as well as in intensity and quantity. Additionally, we know that the underlying neurobiological processes are especially related to the (limbic) motivational and reward systems. These systems are stress-sensitive and, in principle, changeable [12]. This begs the question that is central to today’s happiness research: Is happiness a fixed “factory setting”, i.e., a trait with which one is born or equipped in early childhood? Does an innate happiness set point exist? Alternatively, is happiness learnable and trainable, i.e., dynamic, flexible, and thus changeable? Additionally, if that were the case: Which circumstances and techniques are particularly “effective” in changing happiness and its trajectories?

The question of whether happiness—including its three stages, the U-curve—is genetically determined or innate, at least established (set) in early childhood, i.e., imprinted in our “hardware”, or whether it is constantly and dynamically changeable and yet “designable”, adapting to a certain extent to the living conditions—this question still remains to be answered. Both aspects seem to play a role. However, it is now generally assumed that a significant part of the function of the reward system is individually determined, e.g., by the different genotypes of enzymes tasked with the formation and breakdown of reward hormones and neurotransmitters, the varying receptors and different densities of neurons in the brain’s reward system, etc. However, possibly up to 60% of an individual’s experience of happiness and life satisfaction can be attributed to a fundamental changeability and behavior or learning (c.f., [109,110,111]).

Present studies conducted in our research lab seek to assess how trauma, crisis, or simply getting older, processing and learning from life experiences in general, can change the processes or “setpoints” in the reward system—and thus influence happiness, satisfaction and contentment. In this context, it is important to note that, e.g., for different meditation and mindfulness techniques, and for contemplative practice in general, an influence on the reward and motivation systems or a correlation with their function has been proven [69,73,74,112,113,114]. Long-term effects of contemplative practice have also been demonstrated, not only in relation to alterations in the limbic system, including the amygdala, insular cortex, anterior cingulate cortex, etc., but also to the connectivity between these areas [112,114,115,116]—in addition to areas for empathy, altruism and self-reference [68,112,117,118,119,120]. All of the areas mentioned are highly relevant to our considerations: Basically, meditation techniques, and mindfulness in particular, can influence essential social and emotional-affective functions, which in turn are connected to the reward system—and hence to the experience of happiness. Thus, contemplative practices may be seen—and therefore used—as a means to demonstrate and paradigmatically research the changeability and plasticity of the reward and motivation systems and other relevant CNS areas, i.e., “learned happiness”. In fact, particularly the C-state of happiness (non-wanting, including affiliation/connectedness and altruism/generosity, “inner peace”, etc.; see above) almost seems to equal the notion of deep meditative states (cf. [7]).

Various studies have shown that personality traits that are associated with a high level of life satisfaction include aspects of faith, i.e., religiosity or belief in a higher power, of giving and letting go, of love and loving relationships or, in particular, of a dedication in relation to one’s own actions and work (commitment, flow, mindfulness) [89,90,121,122,123,124]. Indeed, this is where contemplative practices come in. Common concepts of health care and health promotion, of resilience and hardiness, as well as stress reduction in principle are based on contemplative, meditative or mindfulness practices [12,16,17,124,125,126]. Depending on what program is utilized, social connectedness remains a central element, as well as the question of motivation and belief, or membership in a religious community [90,127,128,129].

Healthy aging as well as successful stress management, i.e., coping with challenges (overall a favorable way of dealing with stress [12,71,108,125]), therefore, not only seems to improve health in general and increase life expectancy [7,12,71,108,130,131,132,133,134], but it may be precisely these effects that can be increased through contemplative practice (see above; also [135,136,137]). Hence, the neuronal structures involved in this as well as respective neuroendocrine signaling pathways can be demonstrably influenced by regular meditation practice [12,69,74].

In summary, a generally increased level of life satisfaction and happiness can be assumed in the context of regular meditation or contemplative practice, also referring to the experience of inner growth and “transcendence” that often accompany these practices, which may presumably affect the course of happiness over the life span, as described. Ultimately, however, further studies are necessary and are currently being conducted in our lab.

## 5. Discussion

Biology has equipped humans well to deal with the inevitable changes, ongoing stress and challenges that we encounter throughout our lives. The core aspect of this ability is the so-called auto- or self-regulation [16,17,108,138]. Self-regulation works via the brain’s reward and motivation systems, as does the placebo effect [5,12,68,72,112,139,140,141,142,143], i.e., self-regulation processes can facilitate health promotion [16,17,108,138]. This potential has also been termed “salutogenesis” [144,145].

Humans have a body, a mind, and a brain that allows us to cope with challenges by rewarding beneficial behaviors, or by motivating stimuli that potentially elicit learning curves and personal growth throughout life. This can make us feel happy, and content, over time. For example, social connectivity is rewarded, or trust and belief, finding or having meaning in life, having a spiritual or cultural place and some kind of rootedness, a true home [9,10,146,147]. Here, contemplative practice, mindfulness and meditation can truly be beneficial.

Regarding the neurobiological framework for this, as shown above, three different motivational (sub-) systems exist in the human brain: an incentive-focused system of excitement, consuming, and achieving that also includes desire, euphoric states and is driven by mesolimbic dopamine signaling pathways (→the wanting system); a threat-focused system of stress aversion—alarm, fear or anxiety, and their avoidance, i.e., safety-seeking—where the rewarding feeling of relief is experienced, connected to the amygdala and involving the ability to effectively regulate stress, depending on physiological stress hormone signaling (→the avoidance system); an affiliation-focused system that relates to caring, soothing, calming, feelings of compassion, connectedness, and love (mother–child-like), i.e., quiescence and safety, which involves endogenous opiate and oxytocin signaling (→the non-wanting system).

Research has shown that these systems may all motivate prosocial behavior, i.e., inter- and intrapersonal growth, and elicit feelings of happiness, satisfaction and contentedness [7,8,148,149,150]. Although all three systems may have different foci regarding their activities throughout the course of life (i.e., being predominant at different times), they also act in combination with each other, i.e., interdependently:

During our youth, we need to aspire, seek pleasure, thrills, and engage in risk-taking behaviors—to be creative, learn, and find solutions to problems. Later in life, accelerated stress and the need to protect what had been acquired (i.e., learned and created) prevails—we need to escape threatening stressors in order to “survive”, and to continue. During this time period, particularly in the middle-aged period, we exercise greater caution in order to avoid personal costs associated with poor decisions. Our primary goal is to protect our lives and those of our offspring, as well as the lives of other relatives. The elderly, however, may now have become experts in recalibrating and calming-down again: Individuals above the age of 55 who are currently in the second (“better”) half of life [9,10,86] may no longer need to engage in constant pleasure-seeking activities to experience ecstasy, nor may they actively attempt to overcome the difficulties and to avoid hardships that were part of their previous lives. In fact, the later part of life may reflect the point when the euphoria experienced during youth has transcended into a profound, comforting, and lasting feeling of happiness and contentedness.

Hence, the aforementioned U-curve, which represents the measurable trajectory of happiness throughout our lives, seems to be more pronounced in men than in women [9,10,151]—and it still remains controversial. The main criticism is the idea of a “survival bias”: Could it be that unhealthy, unhappy people die early, thus leading to an overrepresentation of happy individuals and skewing the curve as a result?

Yes and no. Three facts contradict the assumption of a survival bias as the foundation of the U-curve of happiness:

*First*, there is the premise that happiness and health are directly and linearly correlated with one another. Researchers, ourselves included, have found that this is not entirely true. As one progresses in age, the correlation between health and happiness weakens, allowing us to frame the term of a happiness/satisfaction paradox (see above). Statistically speaking, happiness/satisfaction measures “emancipate” from health measures as one progresses in age, suggesting that happiness seems to become more independent of good health and the idea of a preserved physical integrity (c.f, [9,10,151]).

*Second*, a dip in the happiness curve due to an assumed “survival of the happiest”, that is, the assumption of a multitude of unhappy people dying earlier (thus causing a “valley of tears”, i.e., a low point in the curve of happiness, which in fact can be observed at the age between 40 and 50), would imply a surge in mortality at this exact age, with numbers recovering afterwards (otherwise the data would not depict a U-curve). This certainly is not true—it already contradicts mere experience, as one could easily dismiss this point by basic empiricism: As of today, only a very small fraction of people in the “Western” civilization die in midlife between the ages of 40 and 50. Based on more elaborate calculations [152,153], it has been estimated that even at the age of 75 the possible effect of a survival bias would be smaller than three percent on the satisfaction scale [154], which leads us to the last point:

*Third*, even when subjected to the scrutiny of the application of large-scale mortality tables, the U-curve remains robust [153]. Additional comprehensive analyses further validate these findings in 145 countries around the world, regardless of their political or socioeconomic status or development, thus confirming the existence of the U-curve [92]. However, in order to understand the implications of these calculations, and the subsequent evidence for the U-curve, it is necessary to remember that happiness and subjective well-being (→SWB) need to be viewed as (neuro-) biological states, that is, they are emotional-affective conditions—subjective (“eudemonic” [155,156,157]) perceptions rather than cognitive constructs or “pseudo-objectifying” conceptualizations, an appraisal of one’s life. This perspective also forms the basis for the previously described satisfaction paradox: Despite objective reasons for assuming unhappiness and discontent, paradoxically, higher subjective satisfaction and well-being are reported by the elderly. This premise, i.e., SWB being subjective or non-cognitive by its very nature, is not shared by all authors in the field (c.f., [158,159]).

If we assume, however, that the U-curve reflects actual data trends in our populations, the question arises whether similar phenomena exist in other subjective or socio-emotional domains of human experience. It has been shown that empathy, for example, also follows a U-shaped course throughout the life span, with an increase in empathy being observed after the age of 40 [160]. Empathy in this context can be interpreted as “satisfaction with oneself and the world, the other vis-à-vis”, which reminds us of our discourse on happiness and satisfaction. In this regard, our own work has shown that the U-shaped course of happiness/satisfaction, or its increase in the second half of life, is influenced to a lesser degree by financial worries or health as we get older (see above). In the same way, life satisfaction seems to “emancipate” itself more and more from momentary happiness [9,10]. In this respect, an intriguing question remains as to whether the increase in happiness over the course of life is genetic in nature, innate or, above all, shaped in early childhood or by external circumstances—or whether it is correlated with one’s own behavior, meaning that it can be learned and strengthened over time. As shown above, the latter seems to be the case. This, apparently, also applies to empathy and its ability to be trained, especially in the context of the health professions [161].

To reiterate—our own research clearly indicates that the determinants of happiness can be altered, for example, through formal happiness training or programs for stress reduction, behavioral medicine approaches to lifestyle change (e.g., [19,162]). Contemplative, religious, or meditative practices (e.g., mindfulness) also appear to be effective in this context, as has been shown previously [8,12,64,68,69,125,163].

In summary, one can assume that the U-curve of happiness, which is based on the theoretical “ABC model of happiness”, is empirically verifiable, robust in its occurrence, and at the same time may indicate an increase in life satisfaction in the second half of life. Importantly, the U-curve of happiness seems to be widely independent of good health and instead shows indication of being a skill that can be learned and improved over time, meaning that it may be positively influenced by lifestyle modification and behavior change programs, or meditative-contemplative practices. However, rigorous longitudinal-comparative and controlled studies on effect sizes and possible causal relationships as well as comparisons between individually influential factors are necessary to confirm this assumption.

## 6. Strengths, Weaknesses, Future Research

The neurobiological view of happiness presented here and the proposed division into three “happiness categories” are new. Given their novelty, however, these categories—for now—only represent a simplifying model to gain a better yet initial understanding of the different forms of happiness, as it is particularly based on current knowledge of the brain’s reward and motivation systems, including physiological signaling pathways and neurotransmitter releases in areas of relevance. With this approach, however, a bridge to other disciplines—such as sociological or psychological happiness research, or the medical perspective—is also made possible.

A simple, operable template is proposed for this project (Figure 1, Figure 4 and Figure 5). By nature, such models and operationalizations are reductive, which—with the aim of a better understanding, also for the generation of new research questions and hypotheses that then can be examined—are intended to cover the largest area possible, which is never complete. Indeed, such models represent a status quo that can then be confirmed and expanded, e.g., through new insights in the molecular and physiological mechanisms behind it; yet it can, and expectedly will, be fundamentally questioned.

Such new knowledge is also made possible by increasingly precise research methods that are currently being developed. Thus, the status presented here undoubtedly is in a flux. Nevertheless, the proposed model is, to our knowledge, the first to provide such a broad bridge from molecules to behavior (and to happiness, satisfaction or contentment), combining it with a phylo- and ontogenetic perspective—the evolution of happiness over the lifespan (“ontogeny recapitulates phylogeny”)—that also discusses the fundamental question of how happiness can be learned and trained; which would then include an exemplary influence of contemplative practice as well.

Indeed, the term happiness is often used in studies on psychosocial or socioeconomic experiences and outcomes, much more rarely in relation to medical practice or health variables and determinants. In this social or sociological context, the concept of the SWB, as described above, is often used—which, however, rarely measures an emotional state but rather a secondary evaluation or interpretation (appraisal) of one’s own life, mostly retrospectively (cf., among others, [97]). SWB is also used in the context of empirical, large data panels (cf. [164]).

There are few studies on the neurobiological basis of happiness, and practically none on the various sub-forms of happiness (proposed here) and how they are located in the brain, or on the question of changes in happiness over a lifetime.

Primarily, happiness is an emotional-affective state, an immediate and real feeling—not a secondary construct or a downstream or semiotic interpretation of reality. That too, but it comes afterwards. Happiness is actually above all a primary and rewarding experience. This experience is translated from the reward and motivation systems of the CNS into reality—or vice versa—that is to say, for any experience of such positive mood states, paradigmatically, there is always a neurobiological correlate. This means that happiness—theoretically—can always be measured and represented, e.g., in the brain, blood plasma, etc.

In this article, we establish a connection between the various sub-forms of happiness (as well as the overarching concept of happiness as an umbrella term) and its neurobiological correlates—as well as the different manifestations of happiness derived from them, and different ways of measuring them (cf., among others, [7,18,19,20,21,22,23,24,73]). These can be analyses of receptor density or receptor activity in the identified regions of the reward system, direct or indirect evidence of receptor binding of relevant neurotransmitters (see above and Figure 5), evidence of enzymes/enzyme activities for the formation of such transmitters—or their direct evidence (or their metabolites) in body fluids such as liquor, plasma, saliva, sweat, etc. Alternatively, there is imaging, electrographic, or similar proof of activation of relevant areas, especially in the CNS (cf. [7]).

In this context, however, lies a weakness of our approach: As described, knowledge of the various neurobiological aspects of happiness and the three sub-forms presented here is still new (and therefore limited), which also applies to the methods of their measurability. We, therefore, also use detours of detection or reverse the usual detection path: SWB and happiness are usually measured using self-reported measures, such as visual analogue scales (cf. [164]). The inductive method is typical of this primarily empirical approach—that is, models and theories are subsequently developed for the patterns recognized in larger data pools. We, on the other hand, took a different approach and initially proposed a model derived primarily from basic research (see above) in order to then use a kind of probe to search for the assumed pattern in empirical pools (see, e.g., [9,10,86,87,88,89,90]). This deductive approach appears rather unusual, but it has the advantage that one can combine the subjective first-person perspective with the supposedly objective third-person perspective or experimental bases, starting there. This method can unmask patterns and findings that you would not have seen before, because, if you did not have the probe, you might not know what to look for. A major disadvantage of this approach, however, is that you only see what you want to see in the pool (if it exists), i.e., what you were looking for. One can certainly miss other relevant patterns.

In this respect, this paper can be regarded as a conceptualizing basis for future research that examines, deepens, confirms, rejects, or expands the raised aspects in more detail. Such necessary deepening in the time ahead certainly includes more studies on the neuromolecular and physiological foundations of the proposed model, and its various aspects, including the fundamental connection between biology, evolution, rewarding emotional states, and happiness. In addition, more specific, easily accessible and stable biological markers for the individual “happiness categories” (A–C) will also be of great help in the future.

## 7. Conclusions

Happiness is primarily a feeling, not a cognitive construct. Happiness is an immediate experience, and it is based on physiological activity of the brain’s neurobiological reward and motivation systems. This system can be found in traces—in its basic structure—even among the simplest of living beings. It was retained in biological evolution, developed further, but the general principle remained the same (which, in its simplicity, but also quite obviously in its effectiveness, gave reason for biology not to change it essentially).

Today we know three types or subsystems of happiness, i.e., of reward and motivation: (A) wanting and approaching, (B) avoiding and detaching (departing), and finally (C) non-wanting and staying in the here and now. Behind these three types of happiness (ABC) lies a sophisticated neurobiological dynamic and gradation, ranging from the more youthful anticipation, pleasure and ecstasy (A), to stress processing, escape and some kind of relief (B), as we tend to find them in the middle phases of life, until deep satisfaction, quiescence and inner joy (C), which today, as research indicates, is mainly attributed to older people.

The outlined triad of happiness also includes a change in the “timbre” of reward and motivation over lifetime, which leads to the paradoxical finding that, despite objectively existing limitations in well-being and demonstrable reasons for a decrease in quality of life over time, it is indeed the elderly who seem to be the “happiest” (i.e., the most satisfied and contented). This phenomenon has now been termed the “satisfaction paradox”.

The dynamic change in happiness and contentment over the life span, which may also include an increasing “emancipation” from the idea of good health as a mandatory prerequisite for happiness and contentment, is not static or predetermined, but can be acquired and strengthened with practice. In this context, programs and techniques of mindfulness, contemplation, and stress reduction in general, should be mentioned, including those of positive psychology, which seem to be able to influence the life course of happiness and contentment: Happiness can be shaped through practice.

## Figures and Tables

**Figure 1 biology-11-00843-f001:**
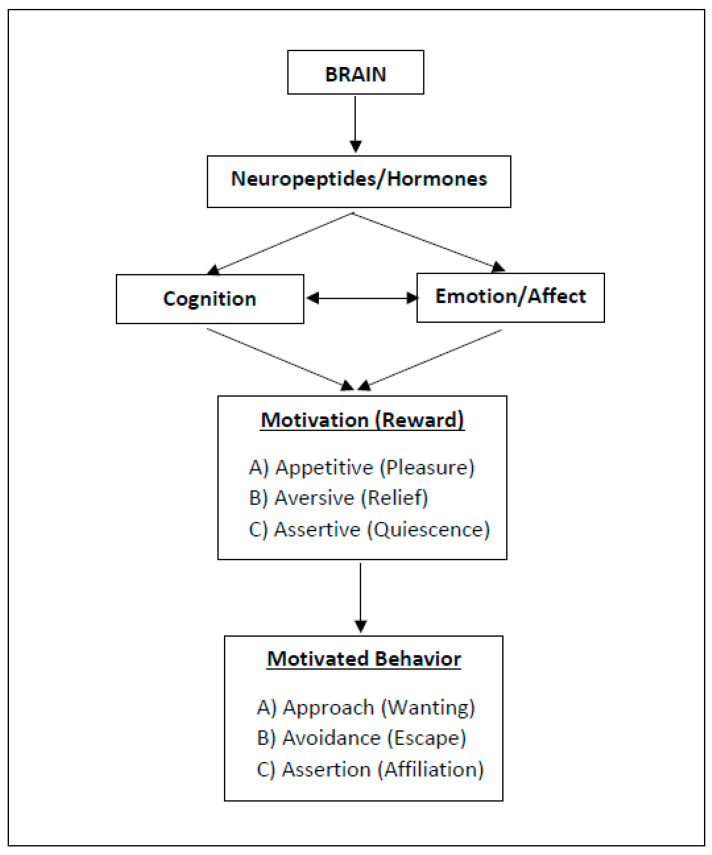
Three Types of Motivation and Reward (ABC Model). Motivational salience is the attribute of a stimulus and can be appetitive, aversive, or assertive. The related reward serves to steer biologically relevant core behaviors (for neurobiological implications, references: see text).

**Figure 2 biology-11-00843-f002:**
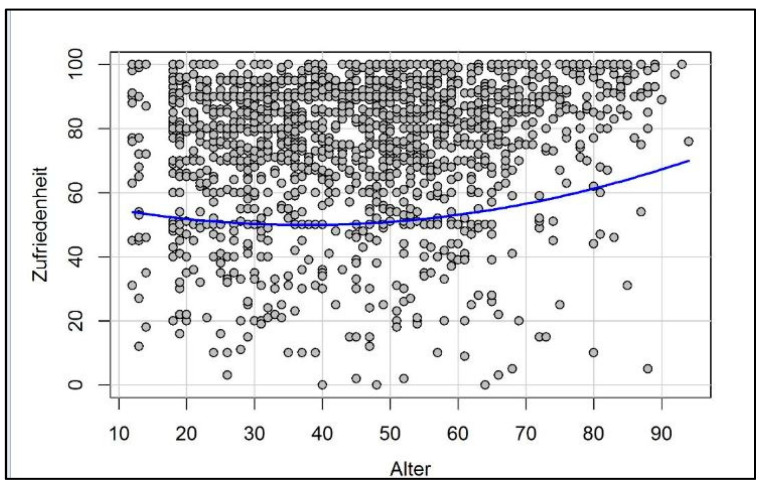
U-Curve of Happiness. Unpublished raw data from our *Experience of Salience and Happiness (ESH)* database depicting a flat U-curve correlation between happiness/life satisfaction (“Zufriedenheit”) and age (“Alter”); cross-sectional correlation analysis: empirical evaluation based on data from 1597 individuals aged 12 to 94 in Germany (for further information, see [9,10]).

**Figure 3 biology-11-00843-f003:**
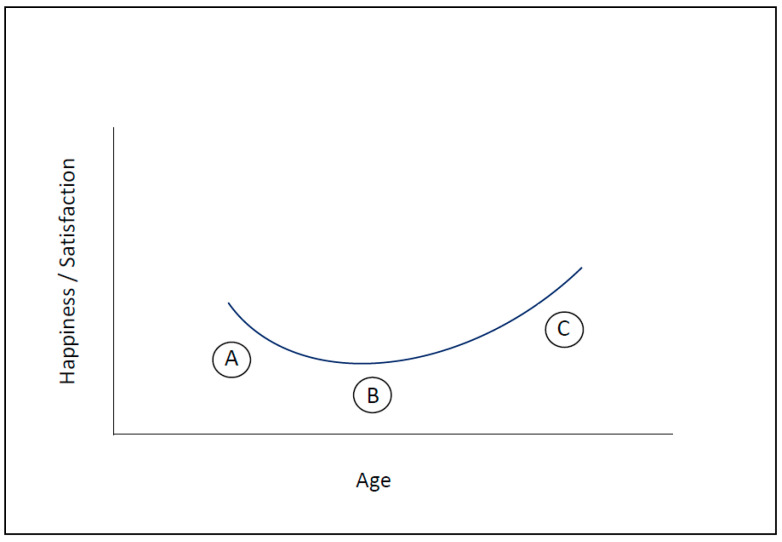
The Three Types of Motivation and Reward (“Happiness ABC”)—Predominance in the Course of Life. In the model, the different types of motivation (ABC—as outlined) predominate in a sequence that follows neurobiological (e.g., development, generativity) and biochemical (e.g., neurotransmitter metabolism) trajectories. Embedded is a movement from “me” to “we”, i.e., individual transcendence over the life span, as well as a transformation from momentary or more instable, fluctuating happiness (c.f., peak moments)—predominant in the young—to more stable or persistent satisfaction levels (c.f., contentedness) in the elderly; this includes becoming more independent of outer/external conditions in favor of internal motives (for further explanations, information: see text).

**Figure 4 biology-11-00843-f004:**
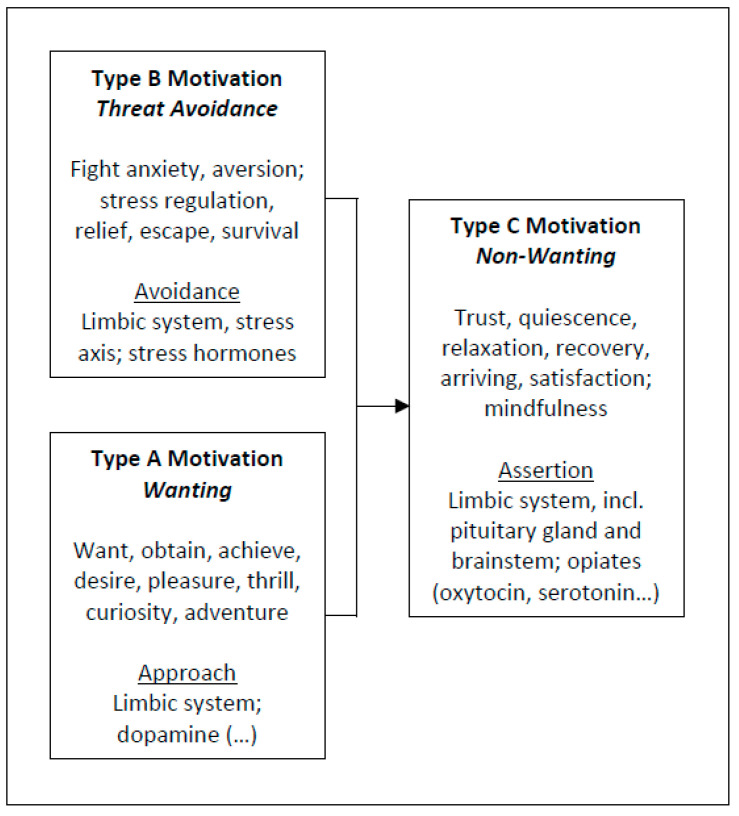
Interplay of ABC Motivations. The different types of motivation (reward) are interwoven in such a way that in our model type A motivation through the mediation of type B increasingly sets in type C, over time—this relationship has recently been established and also includes the neurotransmitters involved, which appeared at different times in evolution and enzymatically emerge from one another, i.e., they are formed from one another. The described sequence also expresses maturation processes (ontogenetic, phylogenetic) over time (for further information, references: see text).

**Figure 5 biology-11-00843-f005:**
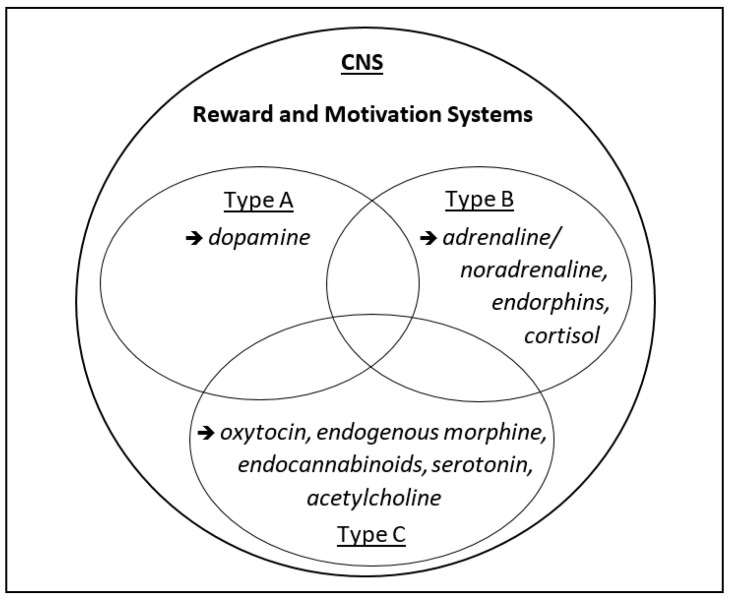
Template for the “Neurobiology of Happiness”. Types A–C of happiness (positive, rewarding mood states) share a common process—related mechanisms that originate/converge on the CNS reward and motivational systems; these include distinctive brain regions and neurotransmitters (i.e., specific reward messengers, among others; for further information, references: see text).

## Data Availability

Not applicable.

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
