# Peer review of "The ABC Model of Happiness—Neurobiological Aspects of Motivation and Positive Mood, and Their Dynamic Changes through Practice, the Course of Life"

_biology, 2022, doi:10.3390/biology11060843_

Round 1
Reviewer 1 Report
The author had made the correction I requested in the references.
Reviewer 2 Report
Thorough revisions
Reviewer 3 Report
Nice article
Reviewer 4 Report
All of my comments were properly addressed. Thank you.
This manuscript is a resubmission of an earlier submission. The following is a list of the peer review reports and author responses from that submission.
Round 1
Reviewer 1 Report
Thank you for the opportunity to review this very interesting paper about neurobiological aspects of happiness. The review is very well written and the only critique I had to make is for Fig 1 and 4 that are not very clear (not as a content, but as a graphical aspect).
In the reference the author is quoting 2 papers nr 10 and 11.
For 10 maybe replace submitted with unpublished manuscript, in review, and for 10 instead of accepted add something la doi or pages, volume.
Reference 130 Neuron should be with italic.
Author Response
To reviewer 1:
Thanks for thoroughly reading the manuscript and your helpful comments and suggestions.
I have taken into account all your comments and made the appropriate corrections, including improving the resolution of the figures and adjusting the mentioned references.
Reviewer 2 Report
Overall, this is a well-written manuscript. The author reviews the literature and then proposes the ABC model of happiness which seems to be a re-conceptulization of a German study with the author describe the U shaped curve of happiness in biological terms. It would be nice for the author to provide some recommendations for future research. For example, to evaluate the proposed ABC model along with some biological markers to futher make the connection between biology and happiness.
Author Response
To reviewer 2:
Thanks for your helpful comments.
As you had suggested, I have now included recommendations for future research, as you find those, e.g., in the new “strengths, weaknesses, future research” section. Furthermore, I have added more papers (i.e., relevant references, resources) on possible biological markers – e.g., on oxytocin as a potential marker of satisfaction (happiness), and its increases with age.
Reviewer 3 Report
Dear author,
It is an interesting article but there are a few things. Please explain how you can measure happiness in the blood because for example the level of serotonin doesn’t equate to happiness. Secondly please define what you mean by elderly Because there is a major difference between for example a 65 or 70-year-old person who is fairly fit and well and might “only” have diabetes Type II or someone who lives in a residential or nursing home, has lost his independence, his wife has died, hardly ever sees his children and most of his friends have died too. Also if you want to know if older people are happier than younger people, what do you actually mean by younger people, then the only way to answer the question properly would be to ask older people if they are now happier than when they were younger etc. Many of them will talk about the good all days simply because those days were better for them. You don’t take into consideration the influence of money/poverty. Please also add strengths and weaknesses of your article.
Also how have you measured the physiology of happiness? In whom did you measure it? And what did you find?
Finally, it’s not clear if your article is a review article or is it a research article because you talk about your research without being specific what you mean by it, or is this simply an opinion article?
Author Response
To reviewer 3:
Thank you for your thorough and very helpful comments. I have considered all:
- I have added certain new passages and references addressing your question of how one can “find” and measure happiness in the body, e.g., in the brain, blood etc. (also referring to your later question on the physiology of happiness): You will find this supplementary content particularly in the newly added section called “Strengths, weaknesses, future research” (chapter 6). Please know that I have derived our model by a deductive approach (starting with an animal model, transferred to humans by integrating/knowledge on the reward and motivation systems as depicted in the article, followed by matching empirical research and, hence, confirming the model. Today, many sources support said relation between, e.g., brain activation patterns (shown, for example, in fMRI neuroimaging studies) and “happy states”, as described. Thus, I have now included more of those studies, e.g., in chapters 1 (introduction) and 6.
- I have added a new paragraph on the definition of “the elderly” and the various (four) ages, i.e., classification of age groups. You will find this new section at the end of chapter 3. In addition, portions of the discussion address issues to which you referred – the question of comparing older to younger people and what makes each group presumably “happy”. In fact, our happiness questionnaires and the data pool we had derived from them (e.g., see Frontiers in Psychology 2001 and 2022) differentiate between “state” and “trait” happiness, i.e., the appraisal of one’s own life (“looking back”), on the one side, versus the actual experience and feeling of happiness on the other, with the latter presenting our primary definition of happiness (as indeed, the secondary or cognitive assessments of “happy states” come neurobiologically second). I have now added our definition of happiness to the introduction (chapter 1).
- You are right that we did not specifically address the influence of money/poverty in this work, since this aspect does not seem to play a primary role for our model and the proposed three stages of happiness. In general, that is, for the motives and drivers of happiness and life satisfaction, however, money and income, besides other factors (such as health), do in fact play a role – yet the influence of money grows weaker when people get older (as long as their basic needs are met), as we have recently shown and published elsewhere (also confirmed by other authors); for further information, please see, for example, our recent papers on the motives of happiness: org/10.3389/fpsyg.2021.777751 and doi.org/10.3389/fpsyg.2022.837638
- Thank you for suggesting adding a section on strengths and weaknesses – which is now included (new chapter 6).
- I have no specified the type of the article (which is a review) in the abstract and the introduction sections (chapter 1).
Reviewer 4 Report
Dear author,
I'll start with an informal note, saying that I certainly enjoyed reading of the manuscript. Particularly inspiring I found idea of applying the theory "Ontogeny recapitulates phylogeny" to the brain neurotransmitters and the explanations of altered stress-reactivity in the middle-age.
Still, before publication, I suggest a major revision of the manuscript.
In brief, the main points of the review need a more structured explanation and reasoning, that in a current version sometimes are between the lines or being rather intuitive. In certain places, a deep analysis of the referenced articles was required to find a relevant data. I explain it below.
- First, the article lacks a clear definition of happiness, and more importantly, do not describe how it was measured in the cited research.
- However:
Lines: 59-60
Happiness can thus be measured (e.g., in the brain, blood) and is associated with biophysiological changes in the body [7,18,19].
I looked through the references and did not find data on how happiness could be measured in the brain o blood. I think such a reference addressing original data would appropriate here.
- Next, assuming that happiness is a neurobiological process, and that it can be measured, the A. B and C brain mechanism should share common processes. But they were not clearly described in the text.
- The theoretical consideration of the appearance of the U-shaped curve is rather vague. This is one of the key points of the article and, in my opinion, it would be good to present it in more detail.
- No criticism on the researches that supposed U-shape curve of lifespan happiness is provided in the paper. But it still exists.
- Starting from line 276 the evolutionary explanations of the motivation types are provided. It is far not clear for the C-type motivation. It’s mentioned in the conclusion but in a quite implicit manner.
- Chapter 4 originally looks unconnected to the previous ones. The connections become clear just in conclusion.
- Neurobiological aspects are discussed poorly then expected, especially in the terms of providing relevant experimental or clinical data.
- Finally, when providing a new model, one would expect a suggestion or proposal of a research that could support the theory or reject it.
In my opinion, these points should to be properly addressed before publishing the present review in Biology.
Thank you again.
Author Response
To reviewer 4:
Thank you for your thorough and very helpful comments. I have tried to address them all:
- A) I have now included a definition of happiness into the introduction section (chapter 1).
- B) The question of how happiness can be/is measured in the cited research has now been addressed more comprehensively, e.g., in the new section called “Strengths, weaknesses, future research” (new chapter 6). Please know that I have derived our model by a deductive approach (starting with an animal model, transferred to humans by integrating/knowledge on the reward and motivation systems as depicted in the article, followed by matching empirical research and, hence, confirming the model). Today, many sources support the relation between, e.g., brain activation patterns (as shown, for example, in fMRI neuroimaging studies) and “happy states”, as described here. Accordingly, I have now included more of those studies and original data, e.g., in chapters 1 (introduction) and 6 (strengths, weaknesses, future research).
- Please also see my previous answer. More sources are provided now. Indeed, the three happiness states, as described, and leading neurotransmitters for each, linked to the reward and motivation systems, seem to correlate with A) pleasure/thrill/positive expectation (e.g., dopamine), B) stress/relief (e.g., stress hormones), C) altruism, relatedness, quiescence (e.g., oxytocin, serotonin); for these states, additional original data (including more neuroimaging reports) are yet included.
- Thank you for explicitly referring to the question of a shared “common process” between the depicted “ABC” brain mechanisms. I absolutely agree, and so have now added more material to the new chapter 6, yet also included a new figure (Fig. 5) to underline said commonalities – with the reward system being the common denominator.
- The U-curve: I have derived our model by a deductive approach. Thus, we have started with the theoretical considerations, and then the actual model, yet already expecting some kind of U-curve (in empirical data, once these existed), or at least a deviation between age (possibly: health; that is, objective measures), and the subjective quality of life/life satisfaction/well-being (subjective measures), when people get older; we then found our model confirmed in large data sets (e.g., the Harvard Nurses Health Study, the UK Million Women Study, etc.), later then in our own studies (for example, see org/10.3389/fpsyg.2021.777751 ; doi.org/10.3389/fpsyg.2022.837638). We consider this a rather robust logic; however, inferring or inductive approaches need to confirm our findings: As said, first large studies and publications are out (I have cited them, also included more sources and described above methodology, i.e, in more detail in the manuscript now), however, the future will tell what needs to be revised from our model. I have now also included this “relativization” statement into the new “strengths, weaknesses, future research” section (chapter 6).
- Please also see my previous response. More content has been added. However, the discussion (chapter 5) includes an elaborate section on the main criticism of the U-curve, that is, the idea of a “survival bias”. I hope, now, that I can meet this criticism substantively. Furthermore, more reflections on differences between happiness as a secondary/cognitive construct (appraisal, etc.) or a primary (direct) emotion have been added. This also refers to the construct of a “satisfaction paradox” – which is described in detail in the text.
- A new section on the evolutionary explanations of the motivation types (particularly regarding type C) has been added to chapter 3. In addition, more relevant references on the neurobiological evidences are now included (please also see my previous answers).
- Chapter 4, I hope, is now better linked to the overall scope and text – I have added a new section to this chapter that relates our model, and particularly the happiness/motivation type C in it, to contemplative practice. Further relevant references to state that link have also been added. Moreover, I have made more explicit that contemplative practice can serve as an example to show the “plasticity” and “trainability” of happiness and motivation. However, the special issue that I have submitted this review to requires a link to contemplative practice, which is another reason why I generally still left the chapter in the review.
- More neurobiological content and explanations have been added, including a new figure (Fig. 5). With regard to clinical data, again, please know that I have derived our model primarily by a deductive approach (clinical data coming second). Please also see my responses no. 1-4 and 6. In addition, more references on experimental data have now been added.
- A new section on strengths and weaknesses, as well as on suggestions for future research (new chapter 6), has now been added.